# cAmbly: A non-toxic cell-penetrating peptide derived from Amblyomin-X with targeted delivery to mitochondrial and cytoplasmic proteins

**Marcus Vinicius Buri**[1⊙], **Graciana Yokota Garavelli**[1⊙], **Hugo Vigerelli**[1],
**Marcelo Medina de Souza**[1], **Aline Ramos Maia Lobba**[1], **Sonja Ghidelli-Disse**[2],
**Ana Marisa Chudzinski-Tavassi**[1]*

**1** CENTD, Centre of Excellence in New Target Discovery, Instituto Butantan, São Paulo, Brazil **2** Cellzome GmbH, a GSK Company, Heidelberg, Germany

⊙ These authors contributed equally to this work.
* ana.chudzinski@butantan.gov.br

## Abstract

The effective delivery of drugs remains a major challenge in the development of new therapeutic molecules. Several strategies have been employed to address this issue, with cell-penetrating peptides (CPPs) standing out due to their ability to traverse cell membranes with minimal cytotoxicity and their relatively straightforward synthesis when conjugated with other molecules. However, while CPPs can successfully enter the cytoplasm, they often lack specificity for particular organelles, leaving target engagement to the drug itself. In this study, we present cAmbly, a novel CPP derived from the antitumoral protein Amblyomin-X. Our findings demonstrate that cAmbly efficiently internalizes into T98G cells within 30 min of incubation and preferentially colocalizes with mitochondria, exhibiting a clear affinity for mitochondrial proteins. These results suggest that cAmbly could serve as a promising delivery vehicle for mitochondria-targeted drugs, such as BCL-2 or OXPHOS modulators, which are commonly employed in cancer treatment. Furthermore, we identified the C-terminal of cAmbly as the optimal site for conjugating molecules intended for intracellular delivery.

## Introduction

Cellular membranes have evolved to protect cells from the influx of undesired particles, ranging from small charged molecules to pathogens. This protective mechanism creates challenges for delivering certain types of drugs due to the hydrophobic nature and organization of the cytosolic barrier [1]. An even greater challenge arises with drug delivery across the blood-brain barrier (BBB), a complex structure composed of tissues and matrices that shields the brain from external substances. As a result, therapies targeting neuronal diseases and brain cancers remain a significant hurdle for drug development and delivery [2].

To overcome these biological barriers, various strategies have been developed for delivering non-permeant drugs, including chemical drug modification, electroporation, lipid vesicles,

**Data availability statement:** All relevant data are within the manuscript The Supporting Information files will be available at https://doi.org/10.5061/dryad.sbcc2frhs and reviewers can use the URL http://datadryad.org/stash/share/BE6wpl4NNr9JLTT1U4ED-asbzJKHRlJdCtRH5NF9O4w.

**Funding:** This research was funded by Grant 2015/50040-4, Fundação de Amparo à Pesquisa do Estado de São Paulo and GlaxoSmithKline foundation,"Grant 2020/13139-0, Fundação de Amparo à Pesquisa do Estado de São Paulo and GlaxoSmithKline foundation, and Grant number 2013/07467-1 Fundação de Amparo à Pesquisa do Estado de São Paulo - Centro de Toxinas, Resposta-imune e Sinalização Celular, A.M.C.-T. is a recipient of Conselho Nacional de Desenvolvimento Científico e Tecnológico grant number 303197/2017-0 and Fundação Butantan-PQ grants. The funders had no role in study design, data collection and analysis, decision to publish, or preparation of the manuscript.

**Competing interests:** The authors have declared that no competing interests exist

and cell-penetrating peptides (CPPs) [3]. While each of these methods possesses specific advantages and limitations, CPPs have emerged as a particularly promising approach for clinical applications. This potential is evidenced by the growing number of pharmaceutical companies conducting clinical trials utilizing CPPs for both local and systemic drug delivery [4].

CPPs are typically short, non-toxic peptides composed of up to 40 amino acids, capable of traversing cell membranes while carrying a wide array of bioactive molecules. The discovery of the TAT peptide from HIV-1 in 1983 [5], followed by the identification of the Drosophila Antennapedia homeodomain's ability to penetrate cells in 1991 [6], laid the foundation for CPP research. Since then, extensive efforts have led to the identification of a vast number of natural and synthetic CPPs, with a current database cataloging over 1,700 peptides (http://crdd.osdd.net/raghava/cppsite/). Based on their mechanisms of action and bioactive properties, CPPs have been classified into distinct categories [7]. Despite the promise demonstrated by CPPs, and the fact that over 25 clinical trials involving CPP-based therapeutics are currently underway, no CPP-based drug has yet received approval from the US Food and Drug Administration (FDA) [8]. This underscores the ongoing need to identify and develop new CPPs with therapeutic potential.

One promising candidate has emerged from Amblyomin-X, a Kunitz-type protease inhibitor derived from the salivary glands of the tick *Amblyomma sculptum* [9]. Initially identified as a Factor Xa inhibitor, Amblyomin-X has since demonstrated significant potential as a novel antitumor drug [10]. Structurally, Amblyomin-X consists of two distinct regions: a non-permeant Kunitz-type domain located at the N-terminus, which mediates the protein's cytotoxic activity, and a disordered C-terminal tail. The latter, despite lacking structural homology with any known motifs, has been shown to facilitate the intracellular delivery of Amblyomin-X and fluorescent probes into the cell cytoplasm [9]. Notably, the Kunitz domain alone is incapable of entering or killing tumor cells, whereas the C-terminal tail, while non-cytotoxic, is essential for cell penetration.

In this study, we further investigate the potential of the Amblyomin-X C-terminal tail, here named cAmbly, as a novel cell-penetrating peptide. Specifically, we assess its ability to internalize into glioblastoma cells and identify its intracellular targets.

## Materials and methods

### Peptide synthesis and modification

Native and three modified variants of the cAmbly peptide were synthesized by Peptide 2.0 (VA, USA). The sequence of the native cAmbly peptide is as follows:

EEQTHFHFESPKLISFKVQDYWILNDIMKKNLTGISLKSEEEDADSGEID

To enable experimental analyses, the peptides were modified as follows: cAmbly-FITC: The native peptide was conjugated with the fluorophore FITC at its C-terminal. B-cAmbly: The peptide was modified at its N-terminal with a combination of photo-leucine for crosslinking, a polyethylene glycol (PEG-4) spacer, and biotin for labeling and purification. cAmbly-PL: Similar to B-cAmbly, but with the modifications applied at the C-terminal to ensure no loss of function. The design of these peptides was carried out in collaboration with Cellzome (Fig 1). All peptides were resuspended in water at a final concentration of 400 µM for subsequent assays.

### Cell culture

Human glioblastoma T98G cells were cultured in Dulbecco's Modified Eagle Medium (DMEM) with high glucose, supplemented with 10% fetal calf serum (FCS) and 1% streptomycin/ampicillin. Cells were maintained at 37°C in a humidified atmosphere containing 5% $CO_2$.

## MTT assay

To assess cAmbly toxicity and its potential effects on mitochondrial metabolism, an MTT assay was conducted in quadruplicate. Briefly: 1) T98G cells were plated in 96-well plates. 2) Cells were treated with cAmbly at concentrations ranging from 0.125 µM to 2 µM for 24 or 48 h. 3) Following treatment, the supernatant was removed and replaced with fresh medium containing 5 mg/mL MTT. 4) After 3 h of incubation, the supernatant was discarded, and the formazan crystals were solubilized in 100 µL of DMSO. 5) Absorbance was measured at 585 nm using a spectrophotometer. 6) Cell viability was expressed as a percentage, using untreated controls as 100% viability.

## Internalization assay

The internalization assay evaluated the ability of modified cAmbly peptides to penetrate cells and assessed their interaction differences. Cell Seeding: T98G cells were seeded in 96-well plates at 85–90% confluency. Treatment: Cells were treated with 1 µM of either B-cAmbly or cAmbly-PL peptides at time points of 30 min, 1 h, 3 h, and 6 h. Crosslinking: To facilitate crosslinking of the peptides with their cellular targets, the plates were irradiated with 5 J/cm² UV light. Fixation: Cells were fixed with 4% formaldehyde for 30 min at room temperature. Staining: Nuclei were stained with Hoechst at a 1:10,000 dilution and Actin filaments were labeled using Phalloidin-Alexa Fluor 647 at a 1:500 dilution. Modified peptides were labeled with streptavidin-Alexa Fluor 488 at a 1:1,000 dilution. Imaging: Images were acquired using an SP8 confocal microscope (Leica Microsystems, Wetzlar, Germany) with objective lenses 20x/0.75 NA and 63x/1.4 NA and laser excitation wavelengths of 405 nm, 488 nm, 552 nm, and 638 nm were used. Image acquisition was performed using LAS X software (Leica Microsystems). Image Processing and Analysis: Images were deconvolved using Huygens Essential software version 22.04 (Scientific Volume Imaging, The Netherlands; http://svi.nl). Quantification of peptide internalization was conducted using CellProfiler software version 4.2.5 (www.cellprofiler.org).

## Mitochondrial colocalization

An essential aspect of evaluating a CPP is determining its organelle-specific targeting. To assess this, we conducted a colocalization assay by treating cells with cAmbly-FITC and labeling mitochondria using Mitotracker Deep Red at a concentration of 300 nM. Image deconvolution and colocalization analysis were performed using Huygens Essential software version 22.04 (Scientific Volume Imaging, The Netherlands; http://svi.nl). Colocalization was quantified using Manders coefficients (M1 and M2).

## Chemoproteomic target discovery

To identify the molecular targets of cAmbly, we performed a chemoproteomic assay. Briefly, cells were seeded in 100 mm plates to achieve 100% confluency and subsequently treated with native and modified cAmbly peptides at varying concentrations for one hour. For competition assays, the native peptide concentration was fixed at 1 µM, while B-cAmbly and cAmbly-PL concentrations ranged from 0 to 6 µM (in duplicate). UV crosslinking (5 J/cm²) was applied to covalently bind the peptides to their targets. Following crosslinking, cells were lysed, and the cAmbly-target complexes were purified using Streptavidin beads. A series of stringent washes were performed sequentially with DTT, NaCl, SDS, and Urea to ensure the removal of non-specific interactions. The bound proteins were digested directly on the beads by adding trypsin, and the resulting tryptic peptides were labeled with TMT tags, as

per the manufacturer's instructions. The labeled peptides from all conditions were combined and desalted using SDB-XC stage tips (Empore, 3M), following established protocols [11]. Samples were analyzed using a qExactive Plus mass spectrometer coupled to a nanoEasy 1200 chromatograph. Peptide separation was carried out on a C18 column (nanoViper C18, 2 µm, 75 µm × 15 cm, Thermo Scientific) under a flow rate of 300 nL/min at 400 bar, using a linear gradient of 5-60% solvent B (80% acetonitrile, 0.1% formic acid) over 90 min. Peptides were detected in data-dependent acquisition mode under positive electrospray ionization conditions. Full-scan peptide signals were acquired within the 300-2000 m/z range at a resolution of 70,000, with an intensity threshold of 1e4. Ions were filtered for fragmentation via the Quadrupole (200-2000 m/z transmission window), and the 10 most intense ions were subjected to HCD fragmentation with a normalized collision energy of 30. MS2 fragments were analyzed in the Orbitrap at a resolution of 17,500, with a cycle time of 40 s determining the number of MS2 events between full scans. Injection parameters were set at 3e6 ions (MS1, 50 ms accumulation time) and 1e5 ions (MS2, 100 ms accumulation time). Protein identification and quantification were performed using MaxQuant software [12], with trypsin specified as the cleavage enzyme and human protein sequences sourced from the Uniprot database. Identifications were accepted at a false discovery rate (FDR) of ≤ 0.1% at both peptide and protein levels, requiring at least one unique peptide match. Mass tolerance thresholds were set at 20 ppm for precursor ions and 40 ppm for fragments. Methionine oxidation and N-terminal acetylation were designated as variable modifications, while cysteine carbamidomethylation was fixed. A maximum of two missed cleavages were permitted. Protein quantification utilized the label-free quantification (LFQ) algorithm in MaxQuant, which normalizes chromatographic peak integration areas for accurate comparisons across samples.

## Results

### cAmbly peptide treatment does not induce cell death in T98G cells

To ensure that cAmbly, as a potential cell-penetrating peptide, does not induce cytotoxic effects, we performed an MTT assay to assess cell viability. T98G cells were treated for 24 h or 48 h with DMEM as the control, DMSO as the solvent control, and cAmbly at concentrations of 0.125 µM, 0.25 µM, 1 µM, and 2 µM. Results were expressed as normalized cell viability ratios, with the controls representing 100% viability. As shown in the 24-hour and 48-hour plots (Fig 2), cAmbly did not reduce cell viability at any of the tested concentrations. Interestingly, cells treated with cAmbly exhibited an increased activity of reducing enzymes, as indicated by viability bars that surpassed those of the controls.

### Time and position of fluorophore labeling interfere with cAmbly internalization

To investigate the internalization and delivery properties of cAmbly, we utilized peptides modified with Biotin-PEG-photoLeucine at either the N-terminal (B-cAmbly) or C-terminal (cAmbly-PL). Cells were treated with these peptides for 0.5, 1, 3, or 6 h, followed by UV crosslinking, paraformaldehyde (PFA) fixation, and labeling with Hoechst (nucleus, blue), phalloidin-647 (actin, red), and streptavidin-488 (peptide, green). Confocal microscopy was employed to ensure accurate three-dimensional localization of the peptides.

Quantitative analysis (Fig 3) and representative confocal images (Figs 4 and 5) demonstrate that both peptide variants successfully entered cells within 30 min of treatment, with peak internalization observed at 1 h and 6 h. Notably, the C-terminal modified peptide (cAmbly-PL) showed a significantly higher intracellular presence compared to the N-terminal modified peptide (B-cAmbly).

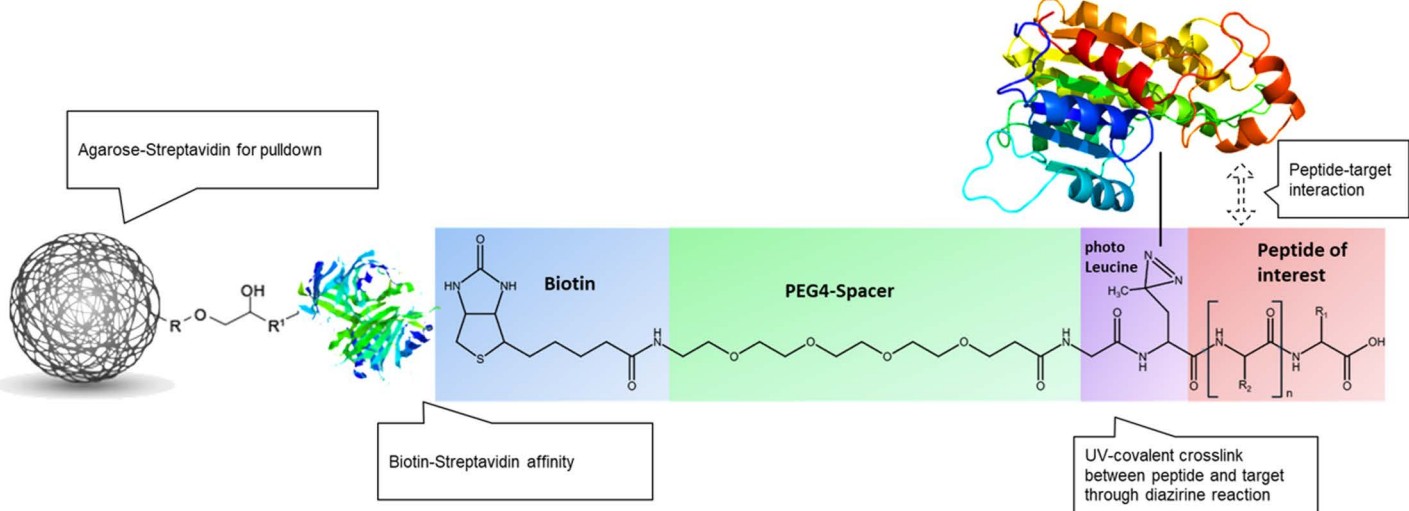

**Fig 1. Molecular schematic of peptide modification and molecular interaction.** The peptide cAmbly (shown in red) is modified with the following components: photo-leucine (purple), which enables covalent binding upon UV excitation; a PEG4 spacer (green) to minimize non-specific interactions; and biotin (blue), which serves as a tag for peptide-target complex pulldown. Agarose-streptavidin beads are employed to capture the peptide-target complex, which is subsequently purified for mass spectrometry analysis.

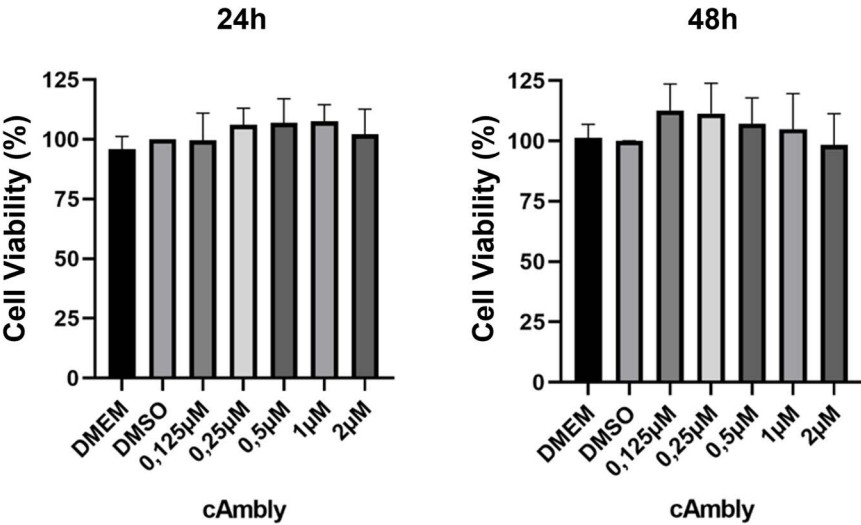

**Fig 2. Cytotoxicity evaluation using the MTT assay.** T98G cells were treated for 24 h or 48 h with varying concentrations of cAmbly (0.125 µM to 2 µM). DMEM and DMSO were used as negative controls. cAmbly did not interfere with cell viability at either time point.

## cAmbly peptide highly colocalizes with mitochondria

To investigate the potential organelle localization of cAmbly, T98G cells were incubated with 1 µM cAmbly-FITC. Mitochondria were stained using MitoTracker Deep Red, and images were acquired via confocal microscopy. A 3D image reconstruction was generated (Fig 6A), and colocalization was quantified using the Mander's Overlap Coefficient (MOC) (Fig 6B) with Huygens Essential software (Scientific Volume Imaging). The MOC generates two coefficients that represent

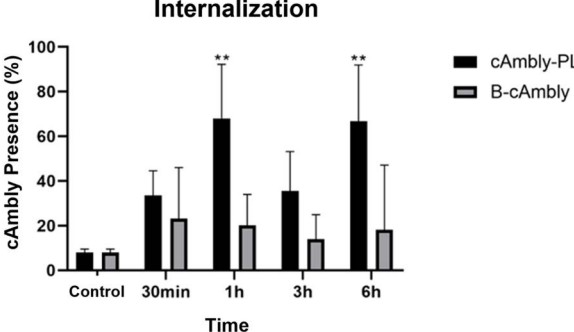

**Fig 3. Quantification of cAmbly internalization.** T98G cells were treated with either N-terminal modified cAmbly (B-cAmbly) or C-terminal modified cAmbly (cAmbly-PL) for 30 min, 1 h, 3 h, or 6 h. Peptide presence was quantified as the percentage of cells containing internalized peptide. cAmbly-PL exhibited robust internalization, with approximately 70% of cells containing the peptide after 1 h and 6 h.

the fraction of colocalizing objects between two channels of a dual-channel image, regardless of intensity. The closer the coefficient value is to 1, the higher the degree of colocalization [13].

The M1 coefficient represents the fraction of red-channel pixels (mitochondria) that colocalize with green-channel pixels (cAmbly), while the M2 coefficient represents the fraction of green-channel pixels (cAmbly) that colocalize with red-channel pixels (mitochondria). The results show that while only a fraction of the stained mitochondria colocalized with cAmbly (M1), most of the cAmbly signal colocalized with mitochondria (M2).

## Chemoproteomic analysis of cAmbly target discovery shows a preference for binding mitochondrial proteins

Identifying potential targets of CPPs after cellular internalization is critical to assess potential off-target effects and ensure the peptide's viability for therapeutic applications or targeted drug delivery. To address this, we employed a chemoproteomic strategy to identify possible cAmbly targets in T98G cells. This approach involves modifying cAmbly to facilitate covalent cross-linking to its targets upon UV excitation, achieved through photo-leucine modification [14]. The peptide-target complex is subsequently purified using streptavidin beads, enabled by biotin modification [15], and identified through mass spectrometry-based proteomics. To ensure comprehensive target identification, cells were treated with a mixture of B-cAmbly (N-terminal modified cAmbly) and cAmbly-PL (C-terminal modified cAmbly), preventing any bias from N- or C-terminal modifications. Additionally, increasing concentrations of native cAmbly (unmodified peptide) were used as a competitor to minimize false-positive target identification. Since native cAmbly lacks both photo-leucine (for covalent crosslinking) and biotin (for pulldown), it competes with the modified peptides (B-cAmbly and cAmbly-PL) for binding sites on target proteins. This competition results in a dose-dependent reduction in the intensity of identified targets, allowing us to distinguish true targets from background signals. We hypothesized that lower concentrations of native cAmbly would result in higher target intensities, while higher concentrations would decrease the target signal due to competitive binding. The first point on the resulting dose-response curves represents background protein intensity caused by TMT (Tandem Mass Tag) imputation analysis, as no modified peptide is available for co-purification at that stage. The profile of protein identification and quantification can be observed at Fig 7.

The proteins exhibiting the most consistent dose-response behavior were: P05165 – Propionyl-CoA carboxylase alpha chain, mitochondrial; P11498 – Pyruvate carboxylase,

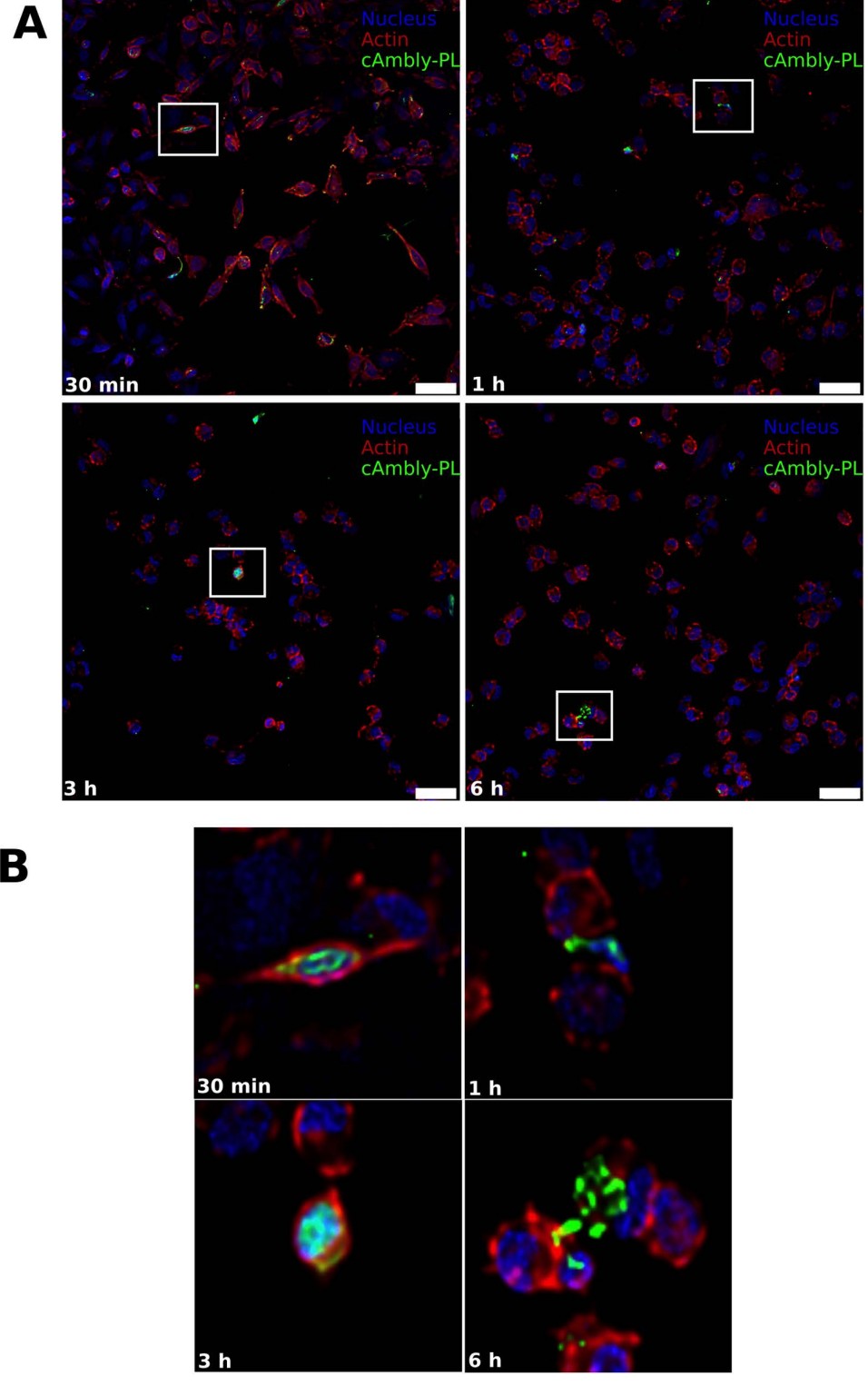

**Fig 4. Internalization of C-terminal modified cAmbly (cAmbly-PL).** T98G cells were treated with cAmbly-PL for 30 min, 1 h, 3 h, or 6 h, followed by crosslinking, PFA fixation, and staining for nuclei, actin, and peptide. **A:** Green arrows indicate cells with internalized peptides. **B:** Enlarged sections provide detailed views of peptide localization at each time point.

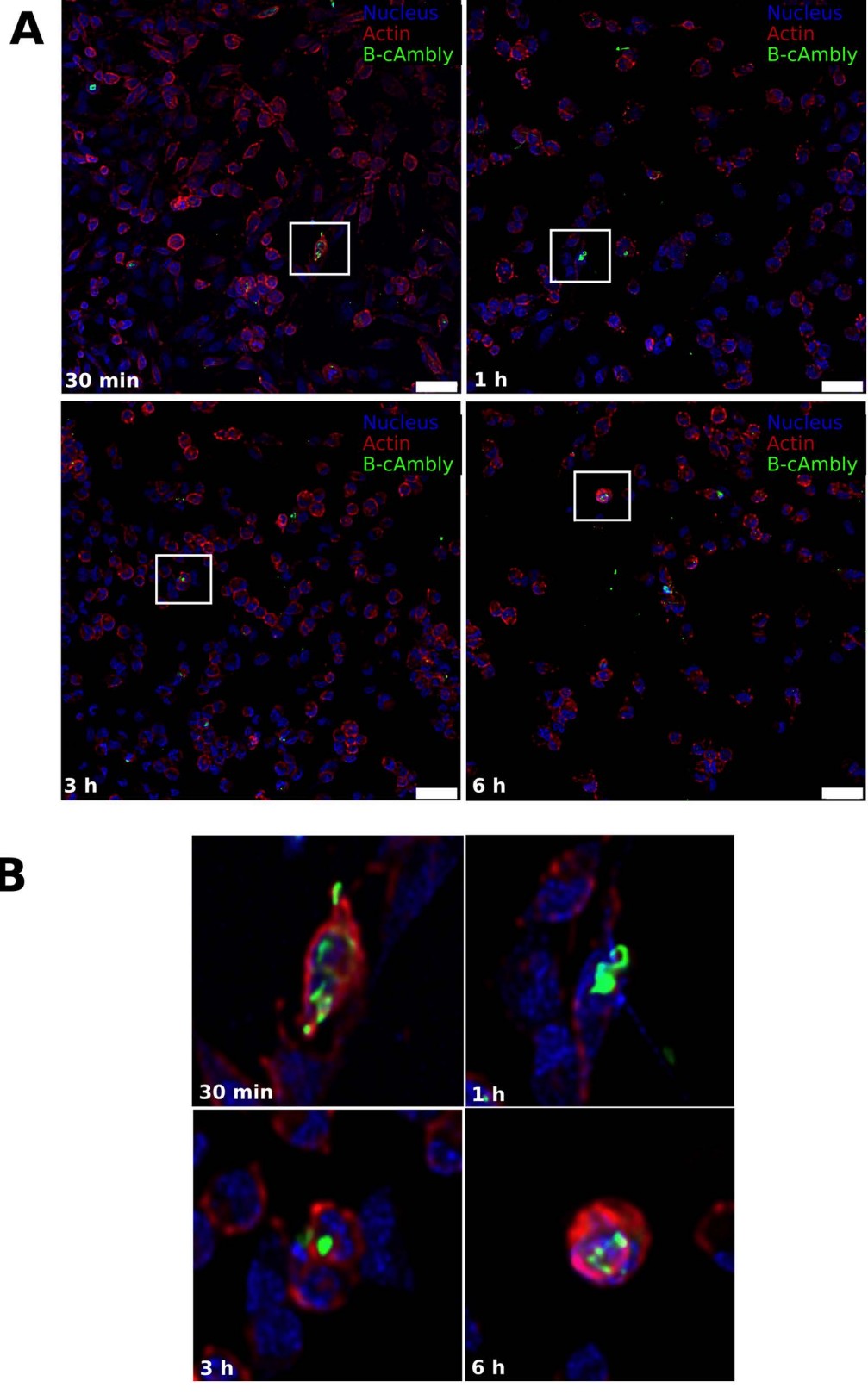

**Fig 5. Internalization of N-terminal modified cAmbly (B-cAmbly).** T98G cells were treated with B-cAmbly for 30 min, 1 h, 3 h, or 6 h, followed by crosslinking, PFA fixation, and staining for nuclei, actin, and peptide. **A:** Green arrows highlight cells containing internalized peptides. **B:** Enlarged sections show detailed peptide distribution within cells at each time point.

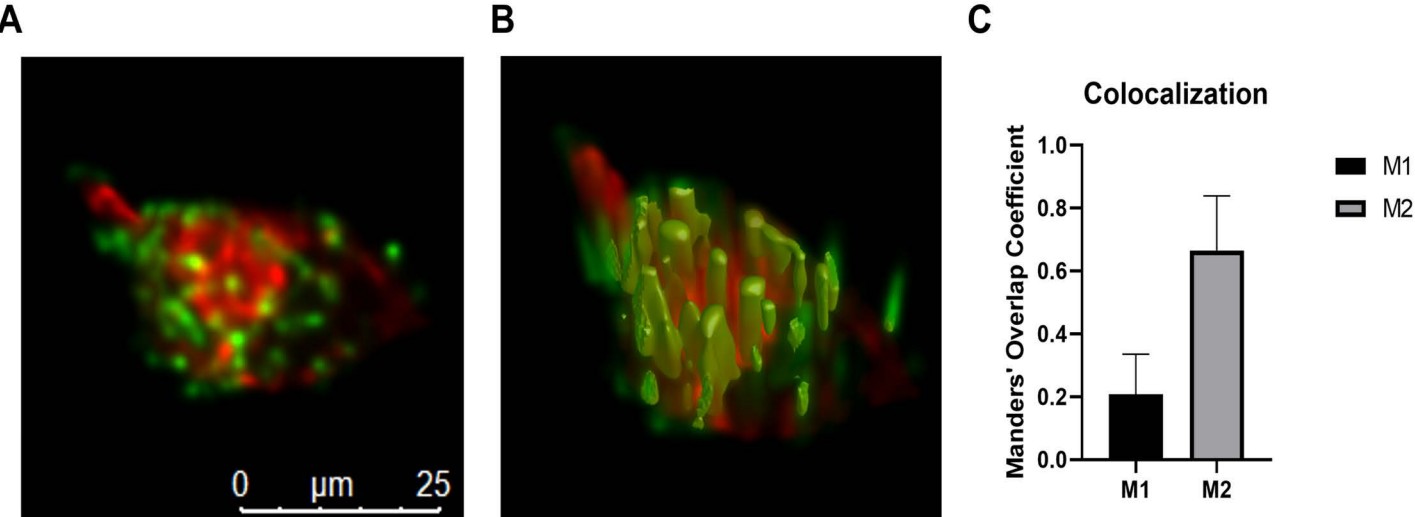

**Fig 6. Colocalization map of cAmbly and mitochondria.** T98G cells were treated with cAmbly-FITC (green), and mitochondria were labeled with MitoTracker Deep Red (red). 2D maximum projection **(A)**, 3D isosurfaces (B) and colocalization quantification (C) were performed using Huygens software. Most cAmbly colocalized with mitochondria (M2), while only a fraction of mitochondria interacted with cAmbly (M1).

mitochondrial; Q9P0W8 – Spermatogenesis-associated protein 7; P60709 – Actin, cytoplasmic 1; P02765 – Alpha-2-HS-glycoprotein; Q96RQ3 – ADP-ribosylation factor-like protein 8A; P69905 – Hemoglobin subunit alpha; Q5T5P2 – Sickle tail protein homolog. Although additional proteins (Table 1) did not display the expected dose-response pattern, they cannot be excluded as potential cAmbly targets.

## Discussion

In this study, we utilized confocal microscopy and chemoproteomic analysis to investigate the cell-penetrating properties of cAmbly, a molecule derived from Amblyomin-X. Our findings provide new insights into the structure-activity relationship of cAmbly and identify potential targets within T98G glioblastoma cells. Using both native and modified forms of the molecule, we demonstrated that cAmbly can efficiently enter cells while carrying conjugated molecules, with preferential localization to the mitochondria and cytoplasm.

Amblyomin-X is a promising antitumoral protein that requires cellular internalization to exert its activity. Previous work by Morais et al. demonstrated that Amblyomin-X induces tumor cell death by inhibiting the proteasome via its Kunitz domain located at the N-terminal region. However, the Kunitz domain alone, without the C-terminal portion, is unable to enter cells or affect them. Conversely, the isolated C-terminal region can penetrate cells but lacks cytotoxic activity [9], highlighting its potential as a CPP.

To confirm the non-toxic nature of cAmbly in glioblastoma cells, we assessed its cytotoxicity against the T98G cell line. Cells were incubated with increasing concentrations of cAmbly (0.125 µM, 0.25 µM, 1 µM, and 2 µM), and viability was measured using the MTT assay. This assay relies on the enzymatic reduction of MTT to formazan by active mitochondria or endosome/lysosome compartments in live cells, a process absent in dead cells [16,17]. None of the tested concentrations significantly reduced cell viability (Fig 2), confirming that cAmbly is non-toxic to T98G cells. Interestingly, while not statistically significant, cells treated with cAmbly showed slightly higher normalized MTT reduction ratios compared to controls, suggesting a potential increase in mitochondrial or lysosomal activity, as these organelles are

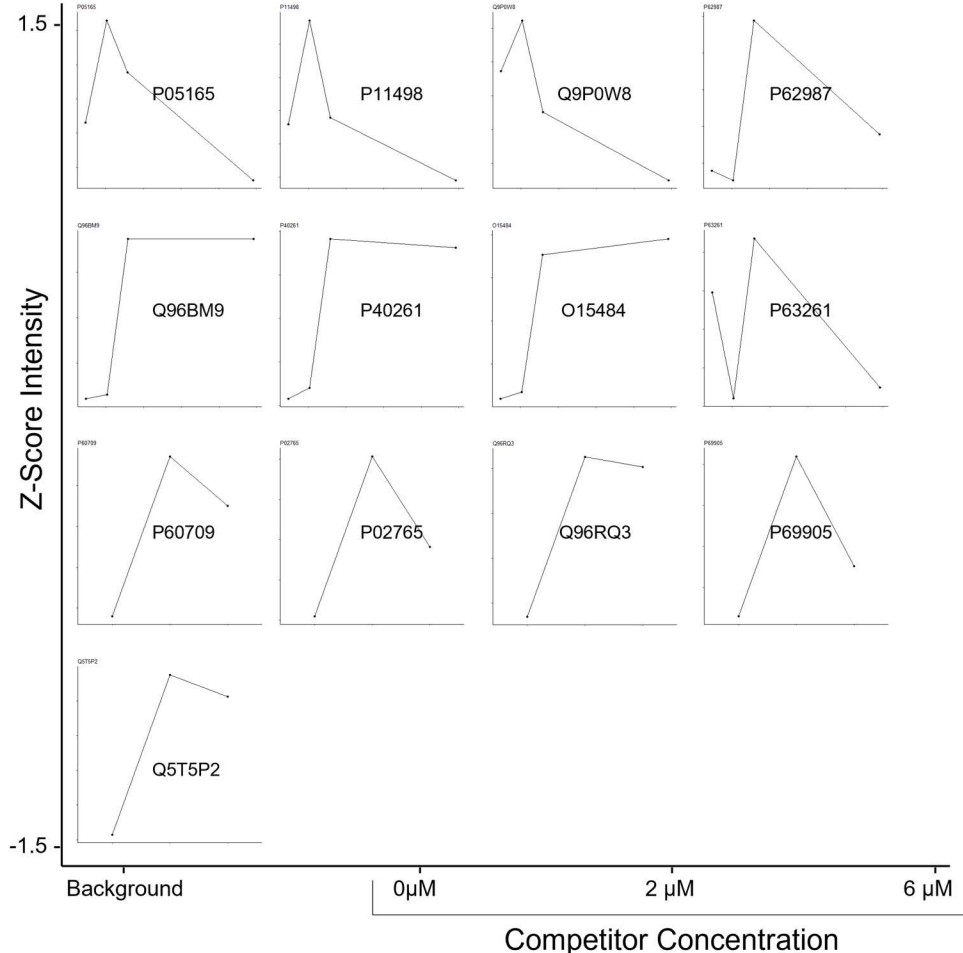

**Fig 7. Dose-response analysis of cAmbly targets.** T98G cells were treated with B-cAmbly and cAmbly-PL in the presence of increasing concentrations of native cAmbly as a competitor. The peptide-target complexes were cross-linked, purified with streptavidin beads, and identified via mass spectrometry. Specific targets exhibit the following pattern: background intensity with native cAmbly alone (first point), high intensity with modified cAmbly alone (second point), and decreasing intensity as the concentration of native cAmbly increases.

responsible for MTT reduction [17]. Although not toxic, the peptide could still trigger an immune response. We observed that the peptide has a preference for mitochondrial proteins and did not identify binding to any immunogenic protein, such as Toll-like receptors. Nonetheless, further experiments on more specialized immune cells, such as monocytes and macrophages, must be conducted to better understand the immunogenicity of cAmbly, evaluating protein binding and cytokine release.

Previous studies have shown that Amblyomin-X internalizes through dynein-mediated transport [18], and its C-terminal region is critical for cellular entry [9]. However, it remains unclear whether cAmbly uses the same pathway for internalization. To optimize cAmbly for drug delivery applications, we introduced modifications at both the N-terminal (B-cAmbly) and C-terminal (cAmbly-PL) regions and evaluated their internalization efficiency. Confocal microscopy revealed that cAmbly could be detected within cells as early as 30 min post-incubation (Figs 3–5). After 1 h and 6 h of incubation, cAmbly-PL was internalized by approximately 70% of the cells, whereas B-cAmbly showed significantly lower internalization efficiency. Interestingly,

**Table 1. UniProt entry, gene name, protein name, and Gene Ontology of identified targets.**

| Entry | Gene Names | Protein names | Gene Ontology |
|---|---|---|---|
| O15484 | CAPN5, NCL3 | Calpain-5 | calcium-dependent cysteine-type endopeptidase activity |
| P02765 | AHSG, FETUA, PRO2743 | Alpha-2-HS-glycoprotein | cysteine-type endopeptidase inhibitor activity |
| P40261 | NNMT | Nicotinamide N-methyltransferase | nicotinamide N-methyltransferase activity |
| P62987 | UBA52, UBCEP2 | Ubiquitin-ribosomal protein eL40 fusion protein (CEP52) | structural constituent of ribosome; ubiquitin protein ligase binding |
| P69905 | HBA1, HBA2 | Hemoglobin subunit alpha | heme binding; oxygen carrier activity |
| Q96BM9 | ARL8A, ARL10B, GIE2 | ADP-ribosylation factor-like protein 8A | alpha-tubulin binding; GTP binding |
| Q5T5P2 | KIAA1217, SKT | Sickle tail protein homolog | – |
| P63261 | ACTG1, ACTG | Actin, cytoplasmic 2 | ATP binding; hydrolase activity; structural constituent of cytoskeleton |
| P60709 | ACTB | Actin, cytoplasmic 1 | ATP binding; hydrolase activity; structural constituent of cytoskeleton |
| Q9P0W8 | SPATA7, HSD3 | Spermatogenesis-associated protein 7 | |
| P05165 | PCCA | Propionyl-CoA carboxylase alpha chain, mitochondrial | ATP binding; biotin binding [GO:0009374]; propionyl-CoA carboxylase activity |
| P11498 | PC | Pyruvate carboxylase, mitochondrial | ATP binding; biotin binding; pyruvate carboxylase activity |
| Q96RQ3 | MCCC1, MCCA | Methylcrotonoyl-CoA carboxylase subunit alpha, mitochondrial | ATP binding; biotin carboxylase activity; methylcrotonoyl-CoA carboxylase activity |

this result was contrary to our initial expectations. Given that the Kunitz domain is located at the N-terminal region of Amblyomin-X, we anticipated that modifications at the N-terminal would enhance internalization. However, our findings indicate that the C-terminal modification significantly improves cellular permeability in T98G cells. This new understanding opens opportunities for further optimization of Amblyomin-X, potentially enhancing its cytotoxic activity by repositioning cAmbly to the N-terminal region of the molecule.

Further evidence indicates that cAmbly preferentially localizes to mitochondria after cellular entry. The slight increase in MTT reduction observed after cAmbly treatment (Fig 2) aligns with the known role of mitochondrial oxidoreductases in MTT metabolism [19]. To confirm mitochondrial targeting, we conducted colocalization experiments using FITC-labeled cAmbly (cAmbly-FITC) and Mitotracker Deep Red to stain mitochondria (Fig 6). The fluorophores were carefully chosen to prevent spectral overlap [20,21], enabling the reliable application of Mander's Overlap Coefficient (MOC) [13]. The small M1 coefficient indicates that only a subset of mitochondria colocalizes with cAmbly, while the high M2 coefficient demonstrates that most cAmbly molecules within the cell localize to these specific mitochondria. This suggests that cAmbly may possess a mitochondrial targeting sequence or an affinity for mitochondrial membranes or proteins.

Our chemoproteomic analysis supports these findings. By combining photo-activated probes with proteomic technologies, we identified potential cAmbly targets in their native cellular environment [22]. To reduce false positives, we employed a competition assay (Fig 7) wherein native peptides compete with photo-leucine-modified cAmbly (Fig 1), for binding to target proteins. Specific targets should exhibit reduced detection in the presence of increasing concentrations of native peptide. The mitochondrial proteins Propionyl-CoA carboxylase alpha chain and Pyruvate carboxylase demonstrated the strongest dose-response behavior.

In addition to mitochondrial proteins, several cytoplasmic proteins, including Actin, Alpha-2-HS-glycoprotein, ADP-ribosylation factor-like protein 8A, Hemoglobin subunit alpha, and Sickle tail protein homolog, also displayed good dose-response behavior. This

indicates that cAmbly may interact with both mitochondrial and cytosolic targets. Of note, Calpain-5, a protein localized to the mitochondria, nucleus, and cytoplasm, also emerged as a target despite showing abnormal dose-response specificity. As Calpain-5 is highly expressed in neuronal cells [23], its interaction with cAmbly may point to a secondary or intermediary role.

It is important to acknowledge potential limitations of the chemoproteomic approach. Both Propionyl-CoA carboxylase alpha chain and Pyruvate carboxylase naturally bind biotin, which raises the possibility of false positives given our use of streptavidin beads for target purification [24]. Nevertheless, the clear dose-response behavior observed for these proteins supports their identification as potential cAmbly targets.

The findings presented here align with the mechanism of action of Amblyomin-X, which induces mitochondrial damage through disruption of the BCL-2 family, leading to reactive oxygen species (ROS) generation and caspase activation in renal carcinoma cells [10]. Our results demonstrate that cAmbly, while not cytotoxic on its own, can efficiently target mitochondria and mitochondrial proteins. This suggests that cAmbly serves as a delivery vector for Amblyomin-X, enabling its localization to mitochondria and subsequent apoptosis induction in tumor cells. Importantly, without the cAmbly portion, Amblyomin-X cannot enter cells and exert its cytotoxic effects [9].

## Conclusion

This study demonstrates the cell-penetrating properties of the cAmbly peptide in T98G cells, highlighting its ability to enter cells without inducing cytotoxicity. Our findings suggest that cAmbly preferentially targets mitochondrial proteins while also interacting with cytoplasmic components. This makes cAmbly a promising candidate for delivering mitochondria-targeted drugs that otherwise struggle to penetrate cells or reach the organelle without causing unintended cytotoxic effects. Importantly, the N-terminal region of cAmbly plays a critical role in its cellular internalization. Therefore, for optimal delivery efficiency, therapeutic molecules intended for mitochondrial targeting could be conjugated to the C-terminal of cAmbly.

## Acknowledgments

*We would like to thank Editage (www.editage.com.br) for English language editing.*

## Author contributions

**Conceptualization:** Marcus Vinicius Buri, Aline Ramos Maia Lobba, Ana Marisa Chudzinski-Tavassi.

**Data curation:** Marcus Vinicius Buri, Hugo Vigerelli, Marcelo Medina de Souza.

**Formal analysis:** Marcus Vinicius Buri, Graciana Yokota Garavelli, Hugo Vigerelli, Marcelo Medina de Souza.

**Investigation:** Marcus Vinicius Buri, Graciana Yokota Garavelli, Hugo Vigerelli.

**Methodology:** Marcus Vinicius Buri, Hugo Vigerelli, Marcelo Medina de Souza, Sonja Ghidelli-Disse.

**Project administration:** Marcus Vinicius Buri.

**Supervision:** Marcus Vinicius Buri, Ana Marisa Chudzinski-Tavassi.

**Writing – original draft:** Marcus Vinicius Buri.

**Writing – review & editing:** Marcus Vinicius Buri.

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
