## [Decision Letter · Decision Letter 0]

23 Apr 2024

PONE-D-23-43062cAmbly, a non-toxic cell-penetrating peptide (CPP) derived from Amblyomin-X, targeting   mitochondrial and cytoplasmic proteinsPLOS ONE

Dear Dr. Chudzinski-Tavassi,

Thank you for submitting your manuscript to PLOS ONE. After careful consideration, we feel that it has merit but does not fully meet PLOS ONE’s publication criteria as it currently stands. Therefore, we invite you to submit a revised version of the manuscript that addresses the points raised during the review process.

We look forward to receiving your revised manuscript.

Kind regards,

Parvez Alam, PhD

Academic Editor

PLOS ONE

 [This research was funded by “Grant 2015/50040-4, São Paulo Research Foundation and GlaxoSmithKline”, “Grant 2020/13139-0, São Paulo Research Foundation and GlaxoSmithKline”, and Grant number 2013/07467-1 (FAPESP-CETICs). A.M.C.-T. is a recipient of CNPq-PQ grant number 303197/2017-0 and Fundação Butantan-PQ grants. R.N.G. is a recipient of FAPESP grant number 2018/20469-7. A.M.A is a recipient of FAPESP grant number 2018/10937-3.].  

[This research was funded by “Grant 2015/50040-4, São Paulo Research Foundation and GlaxoSmithKline”, “Grant 2020/13139-0, São Paulo Research Foundation and GlaxoSmithKline”, and Grant number 2013/07467-1 (FAPESP-CETICs). A.M.C.-T. is a recipient of CNPq-PQ grant number 303197/2017-0 and Fundação Butantan-PQ grants. R.N.G. is a recipient of FAPESP grant number 2018/20469-7. A.M.A is a recipient of FAPESP grant number 2018/10937-3.]

 [This research was funded by “Grant 2015/50040-4, São Paulo Research Foundation and GlaxoSmithKline”, “Grant 2020/13139-0, São Paulo Research Foundation and GlaxoSmithKline”, and Grant number 2013/07467-1 (FAPESP-CETICs). A.M.C.-T. is a recipient of CNPq-PQ grant number 303197/2017-0 and Fundação Butantan-PQ grants. R.N.G. is a recipient of FAPESP grant number 2018/20469-7. A.M.A is a recipient of FAPESP grant number 2018/10937-3.].  

5. We note that your Data Availability Statement is currently as follows: [All relevant data are within the manuscript and its Supporting Information files.]

6. We note that Figure(s) 3, 4,and 5 in your submission contain copyrighted images. All PLOS content is published under the Creative Commons Attribution License (CC BY 4.0), which means that the manuscript, images, and Supporting Information files will be freely available online, and any third party is permitted to access, download, copy, distribute, and use these materials in any way, even commercially, with proper attribution. For more information, see our copyright guidelines: http://journals.plos.org/plosone/s/licenses-and-copyright.

a You may seek permission from the original copyright holder of Figure(s) 3, 4,and 5  to publish the content specifically under the CC BY 4.0 license. 

Additional Editor Comments:

The manuscript by Buri et al., titled "cAmbly, a non-toxic cell-penetrating peptide (CPP) derived from Amblyomin-X, targeting mitochondrial and cytoplasmic proteins," covers a very interesting topic for a wider audience in the field. However, it is not accepted in its current form due to several reasons. It can only be considered for publication after major revision.

1.The manuscript is poorly written and does not follow the flow of information. It should be clearly rewritten, and the rationale for each experiment should be explained very clearly.

2.The purpose of cAmbly CPP targeting mitochondrial proteins should be explained clearly, both in the abstract and introduction. What kind of pathological conditions can it help in targeting, and how?

3.The quality of confocal images is not up to standard and makes it hard to interpret the data. The experiment needs to be repeated or the data should be reanalyzed.

4.Results and discussions should be carefully written to allow readers to understand the importance of this study.

Reviewers' comments:

Reviewer's Responses to Questions

**Comments to the Author**

1. Is the manuscript technically sound, and do the data support the conclusions?

Reviewer #1: Yes

Reviewer #2: No

2. Has the statistical analysis been performed appropriately and rigorously? 

Reviewer #1: Yes

Reviewer #2: No

3. Have the authors made all data underlying the findings in their manuscript fully available?

Reviewer #1: Yes

Reviewer #2: No

4. Is the manuscript presented in an intelligible fashion and written in standard English?

Reviewer #1: Yes

Reviewer #2: No

5. Review Comments to the Author

Reviewer #1: The authors of the manuscript " cAmbly, a non-toxic cell-penetrating peptide (CPP) derived from Amblyomin-X, targeting mitochondrial and cytoplasmic proteins" have used array of methods to demonstrate the cellular internalization of a derived peptide. Overall, the work is very interesting. However, the authors need to address the following concerns:

1.The authors are encouraged to include computational analysis of the designed peptide, encompassing aspects like three-dimensional structure prediction and amino acid parameters. This would enrich the presentation of their work with a comprehensive perspective.

2.The quality of the figures can be improved. Although it can be a pdf conversion issue.

Reviewer #2: The overall manuscript is loosely written the structure and clarity of results and their implications is lacking.

Major points:

1.The abstract provides a good overview of the study however it could benefit from improved clarity and emphasis on the significance of the findings. The abstract should clearly state the significance of the finding for example, colocalizing with mitochondria, with preference for targeting mitochondrial proteins. The importance could be emphasized more. Also,the broader implications of the findings which is use of C-terminal of cAmbly for coupling molecules inside cells should be presented with more clarity and its potential impact on the field.

2.Line 64-66: Clarify which native and modified molecules were synthesized (e.g., specify the precise sequences of each peptide and there extensions in a figure. It may be helpful to provide a brief description of the properties of the native cAmbly sequence.

3.The MTT assay section is clear but it should be supplemented with how cell viability was calculated based on spectrophotometric readings and any controls used in the experiment. Also, consider specifying the number of replicates conducted for each assay to ensure the results' reliability and reproducibility.

4.Consider specifying the solvent control concentration Line 146: Provide more detail on how cell viability was calculated and how normalization was performed. Explain the significance of the lack of cytotoxicity for cAmbly. Line 149-151: The increase in activity of reducing enzymes in cells treated with cAmbly is an interesting finding. However, the implications of this result should be explained further. For instance, why is this increase significant? Does it suggest a potential benefit of cAmbly treatment? The reducing enzymes could be explained more to specify which reducing enzymes and their relevance. Provide more context for the results, particularly in relation to existing literature or expectations. How do these results compare to similar studies or to known effects of other CPPs?

5.Specify whether the UV light exposure is meant to crosslink the peptides with their targets in the cells or on the cell surface. provide information on how long the UV light exposure lasted. Also, consider adding more details about the fixation process (duration and temperature). Mention the concentrations of Hoechst, Phalloidin conjugated to Alexa Fluor 647, and streptavidin conjugated to Alexa Fluor 488 used for labeling. Consider specifying the number of replicates.

6.Mention the concentration of cAmbly-FITC and Mitotracker Deep Red used in the Mitochondrial Colocalization assay. Also, specify the duration of treatment. Provide details on the confocal microscopy imaging conditions, such as settings used for each fluorophore, and how the images were analyzed.

7.Specify the concentrations of native and modified cAmbly peptides used in the assay. Include a description of the controls used in the chemoproteomic assay, such as untreated cells or cells treated with non-modified peptides. Specify the concentration or volume of trypsin used for protein digestion. Provide more information about TMT-tagging, such as the specific reagents or kits used. Clarify the desalinization process using SDB-XC stage-tips (e.g., duration, flow rate). Include more details about the qExactivePlus mass spectrometer settings, including any specific parameters used for data acquisition and analysis. Clarify the methods used for protein quantification, such as specific algorithms or statistical models.

8.It was nowhere mentioned why mitochondria are of interest in this study. Consider stating the reason for focusing on mitochondrial colocalization specifically and why it is important. The lines 193-194 is not clear, present and explain the interpreted results more clearly. Provide more context for the results, such as how these findings compare to existing literature and discuss the implications of the colocalization results for potential therapeutic applications.

9.Chemoproteomic for cAmbly target discovery, explain briefly the mass spectrometry-based proteomics approach used, clarify the methods used for protein quantification. Explain the competitive assay in more detail. For example, describe how the concentrations of non-modified peptides were increased and what the expected outcomes were.

Minor points:

Line 14-15: "The delivery of drugs is a challenge when thinking about new molecules development" could be rephrased for clarity.

Line 16, “outstand this due its capacity" should be revised for clarity. Consider rephrasing to for example: "stand out due to their ability."

Line 25: "Delivery of some types of drugs are not always an easy task" could be improved for clarity. Consider rephrasing for example to "Delivering some types of drugs can be challenging."

Line 37: "small peptides, non-toxic, with up to 40 amino acids" could be improved for clarity. Consider rephrasing for example "small, non-toxic peptides consisting of up to 40 amino acids."

Line 57: "the Kunitz domain could not enter and kill the tumor cell" could gain from being more explicit about the interaction or mechanism involved.

Line 68-70: Explain the rationale for modifying the N-terminal or C-terminal of the peptide and how the design was expected to affect function.

Line 65: Rephrase "one native and three modified molecules of cAmbly peptide" to "one native cAmbly peptide and three modified variants."

Line 69: Rephrase "for purification and labeling purposes" to "for purification and labeling."

Line 73-77: The cell culture section provides the basic conditions for cell growth, also specify the number of cells seeded per well and any other relevant culture conditions.

Line 100: Mention the concentration of cAmbly-FITC and Mitotracker Deep Red used in the assay. Also, specify the duration of treatment.

Line 106: "At 100% confluency" could benefit from specifying the number of cells or density to provide more context on cell seeding.

Line 107: Specify the concentrations of native and modified cAmbly peptides used in the assay.

Line 115: Consider rephrasing "qExactivePlus mass spectrometer coupled to a nanoEasy 1200 chromatograph" for clarity.

6. PLOS authors have the option to publish the peer review history of their article (what does this mean? ). If published, this will include your full peer review and any attached files.

**Do you want your identity to be public for this peer review?** For information about this choice, including consent withdrawal, please see our Privacy Policy .

Reviewer #1: No

Reviewer #2: No

---

## [Author Response · Author response to Decision Letter 0]

26 Jul 2024

June 5th, 2023

PLOS ONE

Dear Editor:

We wish to re-submit an original research article for publication in PLOS ONE, titled “cAmbly, a non-toxic cell-penetrating peptide (CPP) derived from Amblyomin-X, targeting mitochondrial and cytoplasmic proteins.” The paper was co-authored by Marcus V. Buri, Graciana Yokota Garavelli, Hugo Vigerelli, Marcelo M. de Souza, Aline R. M. Lobba, Sonja Ghidelli-Disse, and Ana M. Chudzinski-Tavassi

A recent study showed that the C-terminal portion of Amblyomin-X, an antitumoral protein, was responsible for the internalization of the protein into Tumoral cells. This report presents the potential cell-penetrating properties of the C-terminal of Amblyomin-X, here called cAmbly. We showed that cAmbly was able to internalize T98G cells from 30 minutes of incubation, colocalizing with mitochondria, with a preference for targeting mitochondrial proteins. We also showed that the C-terminal of cAmbly is preferred for coupling molecules to be delivered inside the cells. We showed in this article that even though it is not cytotoxic, targeting mitochondria and mitochondrial proteins cAmbly can deliver drugs to mitochondria, triggering its mechanism of action, for instance, apoptosis.

We would like to answer the major points mentioned by the reviewers:

1. The abstract provides a good overview of the study however it could benefit from improved clarity and emphasis on the significance of the findings. The abstract should clearly state the significance of the finding for example, colocalizing with mitochondria, with preference for targeting mitochondrial proteins. The importance could be emphasized more. Also,the broader implications of the findings which is use of C-terminal of cAmbly for coupling molecules inside cells should be presented with more clarity and its potential impact on the field.

Ans. We agree with the comment, and we added the potential of cAmbly as a cargo delivery for the mitochondria in the abstract and conclusion

2. Line 64-66: Clarify which native and modified molecules were synthesized (e.g., specify the precise sequences of each peptide and there extensions in a figure. It may be helpful to provide a brief description of the properties of the native cAmbly sequence.

Ans. We added a better explanation of peptide synthesis and an image about it

3. The MTT assay section is clear but it should be supplemented with how cell viability was calculated based on spectrophotometric readings and any controls used in the experiment. Also, consider specifying the number of replicates conducted for each assay to ensure the results' reliability and reproducibility.

Ans. We better explained the MTT method

4. Consider specifying the solvent control concentration Line 146: Provide more detail on how cell viability was calculated and how normalization was performed. Explain the significance of the lack of cytotoxicity for cAmbly. Line 149-151: The increase in activity of reducing enzymes in cells treated with cAmbly is an interesting finding. However, the implications of this result should be explained further. For instance, why is this increase significant? Does it suggest a potential benefit of cAmbly treatment? The reducing enzymes could be explained more to specify which reducing enzymes and their relevance. Provide more context for the results, particularly in relation to existing literature or expectations. How do these results compare to similar studies or to known effects of other CPPs?

Ans. We explained in more detail about the MTT experiment. The cAmbly did not affect cell viability, and the slight increase in formazan formation led us to further investigate the effect on mitochondria

5. Specify whether the UV light exposure is meant to crosslink the peptides with their targets in the cells or on the cell surface. provide information on how long the UV light exposure lasted. Also, consider adding more details about the fixation process (duration and temperature). Mention the concentrations of Hoechst, Phalloidin conjugated to Alexa Fluor 647, and streptavidin conjugated to Alexa Fluor 488 used for labeling. Consider specifying the number of replicates.

Ans. The crosslink with 5J/cm2 of UV light is sufficient for covalently binding the peptide with its target through diazirine reaction, wherever the peptide is, inside the cell or at the surface, mimicking interaction with the natural target of the peptide

6. Mention the concentration of cAmbly-FITC and Mitotracker Deep Red used in the Mitochondrial Colocalization assay. Also, specify the duration of treatment. Provide details on the confocal microscopy imaging conditions, such as settings used for each fluorophore and how the images were analyzed.

Ans. All the concentrations were added to the methods section

7. Specify the concentrations of native and modified cAmbly peptides used in the assay. Include a description of the controls used in the chemoproteomic assay, such as untreated cells or cells treated with non-modified peptides. Specify the concentration or volume of trypsin used for protein digestion. Provide more information about TMT-tagging, such as the specific reagents or kits used. Clarify the desalination process using SDB-XC stage-tips (e.g., duration, flow rate). Include more details about the qExactivePlus mass spectrometer settings, including any specific parameters used for data acquisition and analysis. Clarify the methods used for protein quantification, such as specific algorithms or statistical models.

Ans. We better explained the chemoproteomic assay and the competition analysis to include the requested explanations

8. It was nowhere mentioned why mitochondria are of interest in this study. Consider stating the reason for focusing on mitochondrial colocalization specifically and why it is important. The lines 193-194 is not clear, present and explain the interpreted results more clearly. Provide more context for the results, such as how these findings compare to existing literature and discuss the implications of the colocalization results for potential therapeutic applications.

We would like to thank you for this review and added some points about the importance of targeting mitochondria and proteins for cAmbly peptide

9. Chemoproteomic for cAmbly target discovery, explain briefly the mass spectrometry-based proteomics approach used, clarify the methods used for protein quantification. Explain the competitive assay in more detail. For example, describe how the concentrations of non-modified peptides were increased and what the expected outcomes were.

Ans. We better explained the chemoproteomic assay and the competition analysis to include the requested explanations.

10. Thank you for updating your data availability statement. You note that your data are available within the Supporting Information files, but no such files have been included with your submission. At this time we ask that you please upload your minimal data set as a Supporting Information file, or to a public repository such as Figshare or Dryad.

Please also ensure that when you upload your file you include separate captions for your supplementary files at the end of your manuscript.

As soon as you confirm the location of the data underlying your findings, we will be able to proceed with the review of your submission.

Ans. We are uploading the raw files of chemoproteomic assay with the caption at the manuscript

12. When completing the data availability statement of the submission form, you indicated that you will make your data available on acceptance. We strongly recommend all authors decide on a data sharing plan before acceptance, as the process can be lengthy and hold up publication timelines. Please note that, though access restrictions are acceptable now, your entire data will need to be made freely accessible if your manuscript is accepted for publication. This policy applies to all data except where public deposition would breach compliance with the protocol approved by your research ethics board. If you are unable to adhere to our open data policy, please kindly revise your statement to explain your reasoning and we will seek the editor's input on an exemption. Please be assured that, once you have provided your new statement, the assessment of your exemption will not hold up the peer review process.

Ans. All data can be shared after publication without restrictions

13. Please ensure that you include a title page within your main document. We do appreciate that you have a title page document uploaded as a separate file, however, as per our author guidelines (http://journals.plos.org/plosone/s/submission-guidelines#loc-title-page) we do require this to be part of the manuscript file itself and not uploaded separately.

Ans. The title and authors was added along with the manuscript

14. Thank you for stating the following financial disclosure:

[This research was funded by “Grant 2015/50040-4, São Paulo Research Foundation and GlaxoSmithKline”, “Grant 2020/13139-0, São Paulo Research Foundation and GlaxoSmithKline”, and Grant number 2013/07467-1 (FAPESP-CETICs). A.M.C.-T. is a recipient of CNPq-PQ grant number 303197/2017-0 and Fundação Butantan-PQ grants. R.N.G. is a recipient of FAPESP grant number 2018/20469-7. A.M.A is a recipient of FAPESP grant number 2018/10937-3.].

Ans. We added the statement "The funders had no role in study design, data collection and analysis, decision to publish, or preparation of the manuscript."

Other comments: None of our images have copyrights since every art and data were created or generated by ourselves

Further, we would also like to thank the reviewers whose contributions made the article better and more accessible to readers. We want to publish this article in this journal because PlosOne has a broad reach. We believe that the evolution of CPPs can help the development and delivery of drugs against various diseases, being cAmbly a potential candidate for this kind of development, specifically if it is intended to target mitochondria, helping the scientific community solve several health problems.

This manuscript has not been published or presented elsewhere in part or in entirety and is not under consideration by another journal. The appropriate ethics review board approved the study design. We have read and understood your journal’s policies, and we believe that neither the manuscript nor the study violates any of these. There are no conflicts of interest to declare.

Thank you for your consideration. I look forward to hearing from you.

Sincerely,

Ana Marisa Chudzinski-Tavassi

Instituto Butantan, Sao Paulo, SP, Brazil

Phone: +55 11 2627-9555 / email: ana.chudzinski@butantan.gov.br

---

## [Editor Report · Decision Letter 1]

12 Aug 2024

PONE-D-23-43062R1cAmbly, a non-toxic cell-penetrating peptide (CPP) derived from Amblyomin-X, targeting   mitochondrial and cytoplasmic proteinsPLOS ONE

Dear Dr. Chudzinski-Tavassi,

Thank you for submitting your manuscript to PLOS ONE. After careful consideration, we feel that it has merit but does not fully meet PLOS ONE’s publication criteria as it currently stands. Therefore, we invite you to submit a revised version of the manuscript that addresses the points raised during the review process.

**ACADEMIC EDITOR**  Authors have answered most of the reviweres comments. However, quality of figures are still need significant improvement in terms of resolution. It's hard to read figure axis in some places.

We look forward to receiving your revised manuscript.

Kind regards,

Parvez Alam, PhD

Academic Editor

PLOS ONE
---

## [Author Response · Author response to Decision Letter 1]

23 Aug 2024

1. The abstract provides a good overview of the study however it could benefit from improved clarity and emphasis on the significance of the findings. The abstract should clearly state the significance of the finding for example, colocalizing with mitochondria, with preference for targeting mitochondrial proteins. The importance could be emphasized more. Also,the broader implications of the findings which is use of C-terminal of cAmbly for coupling molecules inside cells should be presented with more clarity and its potential impact on the field.

Ans. We agree with the comment, and we added the potential of cAmbly as a cargo delivery for the mitochondria in the abstract and conclusion

2. Line 64-66: Clarify which native and modified molecules were synthesized (e.g., specify the precise sequences of each peptide and there extensions in a figure. It may be helpful to provide a brief description of the properties of the native cAmbly sequence.

Ans. We added a better explanation of peptide synthesis and an image about it

3. The MTT assay section is clear but it should be supplemented with how cell viability was calculated based on spectrophotometric readings and any controls used in the experiment. Also, consider specifying the number of replicates conducted for each assay to ensure the results' reliability and reproducibility.

Ans. We better explained the MTT method

4. Consider specifying the solvent control concentration Line 146: Provide more detail on how cell viability was calculated and how normalization was performed. Explain the significance of the lack of cytotoxicity for cAmbly. Line 149-151: The increase in activity of reducing enzymes in cells treated with cAmbly is an interesting finding. However, the implications of this result should be explained further. For instance, why is this increase significant? Does it suggest a potential benefit of cAmbly treatment? The reducing enzymes could be explained more to specify which reducing enzymes and their relevance. Provide more context for the results, particularly in relation to existing literature or expectations. How do these results compare to similar studies or to known effects of other CPPs?

Ans. We explained in more detail about the MTT experiment. The cAmbly did not affect cell viability, and the slight increase in formazan formation led us to further investigate the effect on mitochondria

5. Specify whether the UV light exposure is meant to crosslink the peptides with their targets in the cells or on the cell surface. provide information on how long the UV light exposure lasted. Also, consider adding more details about the fixation process (duration and temperature). Mention the concentrations of Hoechst, Phalloidin conjugated to Alexa Fluor 647, and streptavidin conjugated to Alexa Fluor 488 used for labeling. Consider specifying the number of replicates.

Ans. The crosslink with 5J/cm2 of UV light is sufficient for covalently binding the peptide with its target through diazirine reaction, wherever the peptide is, inside the cell or at the surface, mimicking interaction with the natural target of the peptide

6. Mention the concentration of cAmbly-FITC and Mitotracker Deep Red used in the Mitochondrial Colocalization assay. Also, specify the duration of treatment. Provide details on the confocal microscopy imaging conditions, such as settings used for each fluorophore and how the images were analyzed.

Ans. All the concentrations were added to the methods section

7. Specify the concentrations of native and modified cAmbly peptides used in the assay. Include a description of the controls used in the chemoproteomic assay, such as untreated cells or cells treated with non-modified peptides. Specify the concentration or volume of trypsin used for protein digestion. Provide more information about TMT-tagging, such as the specific reagents or kits used. Clarify the desalination process using SDB-XC stage-tips (e.g., duration, flow rate). Include more details about the qExactivePlus mass spectrometer settings, including any specific parameters used for data acquisition and analysis. Clarify the methods used for protein quantification, such as specific algorithms or statistical models.

Ans. We better explained the chemoproteomic assay and the competition analysis to include the requested explanations

8. It was nowhere mentioned why mitochondria are of interest in this study. Consider stating the reason for focusing on mitochondrial colocalization specifically and why it is important. The lines 193-194 is not clear, present and explain the interpreted results more clearly. Provide more context for the results, such as how these findings compare to existing literature and discuss the implications of the colocalization results for potential therapeutic applications.

We would like to thank you for this review and added some points about the importance of targeting mitochondria and proteins for cAmbly peptide

9. Chemoproteomic for cAmbly target discovery, explain briefly the mass spectrometry-based proteomics approach used, clarify the methods used for protein quantification. Explain the competitive assay in more detail. For example, describe how the concentrations of non-modified peptides were increased and what the expected outcomes were.

Ans. We better explained the chemoproteomic assay and the competition analysis to include the requested explanations.

10. Thank you for updating your data availability statement. You note that your data are available within the Supporting Information files, but no such files have been included with your submission. At this time we ask that you please upload your minimal data set as a Supporting Information file, or to a public repository such as Figshare or Dryad.

Please also ensure that when you upload your file you include separate captions for your supplementary files at the end of your manuscript.

As soon as you confirm the location of the data underlying your findings, we will be able to proceed with the review of your submission.

Ans. We are uploading the raw files of chemoproteomic assay with the caption at the manuscript

12. When completing the data availability statement of the submission form, you indicated that you will make your data available on acceptance. We strongly recommend all authors decide on a data sharing plan before acceptance, as the process can be lengthy and hold up publication timelines. Please note that, though access restrictions are acceptable now, your entire data will need to be made freely accessible if your manuscript is accepted for publication. This policy applies to all data except where public deposition would breach compliance with the protocol approved by your research ethics board. If you are unable to adhere to our open data policy, please kindly revise your statement to explain your reasoning and we will seek the editor's input on an exemption. Please be assured that, once you have provided your new statement, the assessment of your exemption will not hold up the peer review process.

Ans. All data can be shared after publication without restrictions

13. Please ensure that you include a title page within your main document. We do appreciate that you have a title page document uploaded as a separate file, however, as per our author guidelines (http://journals.plos.org/plosone/s/submission-guidelines#loc-title-page) we do require this to be part of the manuscript file itself and not uploaded separately.

Ans. The title and authors was added along with the manuscript

14. Thank you for stating the following financial disclosure:

[This research was funded by “Grant 2015/50040-4, São Paulo Research Foundation and GlaxoSmithKline”, “Grant 2020/13139-0, São Paulo Research Foundation and GlaxoSmithKline”, and Grant number 2013/07467-1 (FAPESP-CETICs). A.M.C.-T. is a recipient of CNPq-PQ grant number 303197/2017-0 and Fundação Butantan-PQ grants. R.N.G. is a recipient of FAPESP grant number 2018/20469-7. A.M.A is a recipient of FAPESP grant number 2018/10937-3.].

Ans. We added the statement "The funders had no role in study design, data collection and analysis, decision to publish, or preparation of the manuscript."

Other comments: None of our images have copyrights since every art and data were created or generated by ourselves

Further, we would also like to thank the reviewers whose contributions made the article better and more accessible to readers. We want to publish this article in this journal because PlosOne has a broad reach. We believe that the evolution of CPPs can help the development and delivery of drugs against various diseases, being cAmbly a potential candidate for this kind of development, specifically if it is intended to target mitochondria, helping the scientific community solve several health problems.

This manuscript has not been published or presented elsewhere in part or in entirety and is not under consideration by another journal. The appropriate ethics review board approved the study design. We have read and understood your journal’s policies, and we believe that neither the manuscript nor the study violates any of these. There are no conflicts of interest to declare.

ACADEMIC EDITOR: Authors have answered most of the reviweres comments. However, quality of figures are still need significant improvement in terms of resolution. It's hard to read figure axis in some places.

Ans. We concur with the suggestions of the Academic Editor and have made adjustments to the figures to enhance their resolution. Figures 4 and 5 have been replaced with higher-resolution images, and Figure 7 has been redrawn to improve the legibility of the axes.

---

## [Decision Letter · Decision Letter 2]

13 Nov 2024

PONE-D-23-43062R2cAmbly, a non-toxic cell-penetrating peptide (CPP) derived from Amblyomin-X, targeting   mitochondrial and cytoplasmic proteinsPLOS ONE

Dear Dr. Chudzinski-Tavassi,

Thank you for submitting your manuscript to PLOS ONE. After careful consideration, we feel that it has merit but does not fully meet PLOS ONE’s publication criteria as it currently stands. Therefore, we invite you to submit a revised version of the manuscript that addresses the points raised during the review process.

We look forward to receiving your revised manuscript.

Kind regards,

Boyan Grigorov

Academic Editor

PLOS ONE

Journal Requirements:

Reviewers' comments:

Reviewer's Responses to Questions

**Comments to the Author**

1. If the authors have adequately addressed your comments raised in a previous round of review and you feel that this manuscript is now acceptable for publication, you may indicate that here to bypass the “Comments to the Author” section, enter your conflict of interest statement in the “Confidential to Editor” section, and submit your "Accept" recommendation.

Reviewer #3: (No Response)

2. Is the manuscript technically sound, and do the data support the conclusions?

Reviewer #3: Partly

3. Has the statistical analysis been performed appropriately and rigorously? 

Reviewer #3: N/A

4. Have the authors made all data underlying the findings in their manuscript fully available?

Reviewer #3: No

5. Is the manuscript presented in an intelligible fashion and written in standard English?

Reviewer #3: No

6. Review Comments to the Author

Reviewer #3: I have the following comments concerning the manuscript of Marcus Buri et al.:

The manuscript needs to be proofread by a native speaker. Pluriels are missing (among other mistakes), e.g. Chemoproteomic(s), streptavidin bead(s)...

The Results section must be better organized. I was wondering if it were the Materials & Methods section that I was reading. Real explicative subheadings should be added.

Figures should appear in the Results section (actually, Fig.1 appears in Materials & Methods).

Concerning the mitochondria/cAmbly colocalization, include a wide field 2D image

to figure 6 (similar to Fig.4 & 5).

Concerning the discussion section, mention the potential delivery method of the peptides in vivo (systematic or local). What is the immunogenicity of this peptide?

In the conclusion, the authors argue that "cAmbly can potentially deliver mitochondria-targeted drugs...". Can you give examples for such drugs?

7. PLOS authors have the option to publish the peer review history of their article (what does this mean? ). If published, this will include your full peer review and any attached files.

**Do you want your identity to be public for this peer review?** For information about this choice, including consent withdrawal, please see our Privacy Policy .

Reviewer #3: No

---

## [Author Response · Author response to Decision Letter 2]

19 Dec 2024

Thank you for your comments.

The article is now reviewed by professional Editors using Editage Platform.

The supporting files was added using DRYAD platform at doi:10.5061/dryad.sbcc2frhs (URL for the reviewers : (http://datadryad.org/stash/share/BE6wpl4NNr9JLTT1U4ED-asbzJKHRIJdCtRH5NF9O4w

We added a 2D image for Manders’ colocalization analysis (Figure 6)

The scheme that we used at methods was made with the intent to better explain the molecular design we made for the chemoproteomics, and we understand that this figure should be at methods, not at the results

---

## [Editor Report · Decision Letter 3]

7 Jan 2025

PONE-D-23-43062R3cAmbly: A non-toxic cell-penetrating peptide derived from Amblyomin-X with targeted delivery to mitochondrial and cytoplasmic proteinsPLOS ONE

Dear Dr. Chudzinski-Tavassi,

Thank you for submitting your manuscript to PLOS ONE. After careful consideration, we feel that it has merit but does not fully meet PLOS ONE’s publication criteria as it currently stands. Therefore, we invite you to submit a revised version of the manuscript that addresses the points raised during the review process.

The authors should reply and include in the text the following points that have been raised by the reviewers:1. The "Results" must include explicative subheadings (titles) and not simply the method used.2. Discuss about the immonogenicity of the petide and its mode of delivery.3. Moderate the final sentence in the Conclusion: "Therefore, for optimal delivery efficiency, therapeutic molecules intended for mitochondrial targeting COULD be conjugated to the C-terminal of cAmbly." 

We look forward to receiving your revised manuscript.

Kind regards,

Boyan Grigorov

Academic Editor

PLOS ONE
---

## [Author Response · Author response to Decision Letter 3]

8 Jan 2025

Thank you for your comments to make this a better article. Below we are answering each point:

1. The "Results" must include explicative subheadings (titles) and not simply the method used.

Ans.: We changed the subheadings adding more context to the results

2. Discuss about the immonogenicity of the petide and its mode of delivery.

Ans.: At discussion, we added the sentence: “Although not toxic, the peptide could still trigger an immune response. We observed that the peptide has a preference for mitochondrial proteins and did not identify binding to any immunogenic protein, such as Toll-like receptors. Nonetheless, further experiments on more specialized immune cells, such as monocytes and macrophages, must be conducted to better understand the immunogenicity of cAmbly, evaluating protein binding and cytokine release”

3. Moderate the final sentence in the Conclusion: "Therefore, for optimal delivery efficiency, therapeutic molecules intended for mitochondrial targeting COULD be conjugated to the C-terminal of cAmbly."

Ans.: We appreciated the comment and changed the word

---

## [Editor Report · Decision Letter 4]

12 Jan 2025

cAmbly: A non-toxic cell-penetrating peptide derived from Amblyomin-X with targeted delivery to mitochondrial and cytoplasmic proteins

PONE-D-23-43062R4

Dear Dr. Chudzinski-Tavassi,

We’re pleased to inform you that your manuscript has been judged scientifically suitable for publication and will be formally accepted for publication once it meets all outstanding technical requirements.

Kind regards,

Boyan Grigorov

Academic Editor

PLOS ONE
---

## [Editor Report · Acceptance letter]

PONE-D-23-43062R4

PLOS ONE

Dear Dr. Chudzinski-Tavassi,

I'm pleased to inform you that your manuscript has been deemed suitable for publication in PLOS ONE. Congratulations! Your manuscript is now being handed over to our production team.

Kind regards,

on behalf of

Dr. Boyan Grigorov

Academic Editor

PLOS ONE